

**Distribution and Flux of Dissolved Iron of the Rajang and Blackwater**
**Rivers at Sarawak, Borneo**
Xiaohui Zhang[1], Moritz Müller[2], Shan Jiang[1], Ying Wu[1], Xunchi Zhu[3], Aazani Mujahid[4], Zhuoyi Zhu[1],
Mohd Fakharuddin Muhamad[4], Edwin Sien Aun Sia[2], Faddrine Holt Ajon Jang[2], Jing Zhang[1]
[1]State Key Laboratory of Estuarine and Coastal Research, East China Normal University, 200241
Shanghai, China
[2]Swinburne University of Technology, Faculty of Engineering, Computing and Science, 93350
Kuching, Sarawak, Malaysia
[3]School of Ecological and Environmental Sciences, East China Normal University, 200241
Shanghai, China
[4]Faculty of Resource Science & Technology, University Malaysia Sarawak, 94300, Sarawak,
Malaysia
*Correspondence to:* Xiaohui Zhang (52163904020@stu.ecnu.edu.cn)





**Abstract** Dissolved iron (dFe) is essential for biogeochemical reactions in oceans, such as photosynthesis, respiration and nitrogen fixation. Currently, large uncertainties remain on riverine dFe inputs, especially for tropical rivers in Southeast Asia. In the present study, dFe concentrations and distribution along the salinity gradient in the Rajang River in Malaysia, and three blackwater rivers draining from peatlands, including the Maludam River, the Sebuyau River, and the Simunjan River, were determined. In the Rajang River, the concentration of dFe in fresh water (salinity<1) in the wet season (March 2017) was higher than that in the dry season (Auguest 2016), which might be related to the resuspension of sediment particles and soil erosions from cropland in the watershed. In the Rajang Estuary, an intensive removal of dFe in low salinity waters (salinity<15) was observed, likely due to the salt-induced flocculation and the absorption onto suspended particulate matters (SPM). However, dFe concentration enhancements in the wet season occured in some sampling sites, which may be related to the desorption from SPM and agriculture activities. On the other hand, dFe was conservatively distributed in high salinity waters (salinity>15), which may result from the association between dFe and pelagic organic matters. In the blackwater rivers, concentrations of dFe reached 44.2 $\mu$mol L$^{-1}$, indicating a great contribution from peatland. The dFe flux derived from the Rajang Estuary to the South China Sea was $(6.4\pm2.3)\times10^5$ kg yr$^{-1}$. For the blackwater river, the dFe flux was approximately $(1.1\pm0.5)\times10^5$ kg yr$^{-1}$ in the Maludam River. The anthropogenic activities may play an important role in the dFe yield, such as the Serendeng tributary of the Rajang River, and Simunjan River, where intensive oil palm plantations were observed.



## 1. Introduction

Iron (Fe) is an essential element for enzymes and deemed to be responsible for photosynthesis, respiration, and nitrogen fixation (Moore et al., 2009; Raven, 2010; Williams, 1981). In the past four decades, Fe has been identified as micronutrient, significantly supporing primary productivity in oceans (Brand and Sunda, 1983; Moore et al., 2009; Tagliabue et al., 2017). In particular, after a series of *in-situ* fertilization experiments, researchers verified the Fe limitation on the growth of phytoplankton and the critical role in the $CO_2$ fixation (Boyd et al., 2007; de Baar et al., 2005; Martin, 1990).

On a global scale, the riverine dissolved iron (dFe) transported to coastal oceans is estimated to be $1.5 \times 10^9$ mol yr$^{-1}$ (Boyd and Ellwood, 2010; de Baar and de Jong, 2001; Jickells et al., 2005; Milliman and Farnsworth, 2011; Saitoh et al., 2008). Tropical rivers might contribute a significant amount of dFe based on studies from the Amazon River (Bergquist and Boyle, 2006; Gaillardet et al., 1997), and the Congo River (Coynel et al., 2005; Dupré et al., 1996). However, few studies have assessed dFe concentrations and transport in tropical rivers in Southeast Asia, even though those rivers can account for about 30% of fluvial discharge to oceans (Milliman and Farnsworth, 2011).

Estuaries, as the interaction zone between surface water and coastal oceans, could fundamentaly modulate dFe concentrations during the mixing, and hence change the magnitude of riverine dFe flux. There is a large volume of published studies on the behaviors of dFe in a wide range of estuaries (Boyle et al., 1977; Herzog et al., 2017; Oldham et al., 2017; Zhu et al., 2018). In particular, some estuarine environments are enriched with organic matters because of high primary productivity and terrestrial loading, as well as the great contributions from salt marshes and peatlands. These organic matters may deeply affect the distribution of riverine solutes (Hedges et al., 1997; Müller et al., 2015). Generally, estuaries act as a sink for dFe due to the flocculation between the cations and the high molecular colloids (Bergquist and Boyle, 2006; Boyle et al., 1977; Stolpe and Hassellov, 2007). The magnitude of dFe removal in the estuary can be quantified by removal factors (RF). However, in some rivers with high concentrations of dissolved organic matters (DOM), conservative distribution of dFe was found, because of the chemical connection of Fe to DOM (Oldham et al., 2017; Sanders et al., 2015; Stolpe et al., 2010). More importantly, large populations in estuaries are frequently observed. Anthropogenic activities, such as coal mining, ore industry, and agriculture activities, could





significantly impact concentrations and distributions of dFe in estuaries (Braungardt et al., 2003;
Morillo et al.,2005; Xue et al., 2016).
Currently, only limited records on the dFe concentrations were provided in peatland draining rivers
(Batchelli et al., 2010; Krachler et al., 2010; Oldham et al., 2017). The dFe distribution in the peatland
draining estuaries is also largely unknown. Southeast Asia hosts a large area of peatlands along the
coastal belts, with a coverage of approximately 9% on a global scale (Dommain et al., 2011; Joosten,
2012). For dFe research in Malaysia, to the authors' best knowledge, the dFe concentration was only
determined (1) in the fresh water at Pelagus, where the high concentration of dFe was observed,
resulting from sediment diffusion (Siong, 2015); (2) in Bebar, a blackwater river in Pahang, Malaysia,
the concentration of dFe was up to 30 μmol L$^{-1}$, but the information about the distribution and
biogeochemistry of dFe was missing (Shuhaimiothman et al., 2009). Such knowledge limitation may
markedly influence the regional dFe budget estimation.
To fill this gap, two cruises were conducted in Sarawak state, Borneo, Malaysia, including the largest
river in Sarawak State (the Rajang River) and three peat-draining rivers. This study aims to determine
(1) the concentration and distribution of dFe, (2) the seasonal variation of dFe in the Rajang River,
(3) the dFe yields and the magnitude of riverine fluxes to the coastal areas.
**2.  Materials and methods**
**2.1 Study area**
Malaysia has the second largest peatland areas (about $2.6\times10^4$ km$^2$) in Southeast Asia (Mutalib et al.,
1992). Sarawak State accounts for the largest peatland area in Malaysia, and has a wide spread of
blackwater rivers (Joosten, 2012; Wetlands International, 2010). Approximately 23% of the peatland
is defined as relatively undisturbed in Malaysia, in which 17% are in Sarawak (Wetlands International,
2010). Since the mid-1980s, rubber, textiles, metals, food processing, petroleum, and electronics have
been developed, and have become the major economic support in Malaysia (Trade Chakra, 2009). As
a response, deforestation rate in Sarawak increased to 2% yr$^{-1}$ from 1990 to 2010 (Miettinen et al.,
2012), and this rate is attributed to oil and rubber plantations (Joosten, 2012).
The Rajang River, i.e. the largest river in Malaysia, flows from the Iran Mountain to the South China
Sea (Fig. 1a and b), with a length of 530 km. The drainage basin is $5.1\times10^4$ km$^2$ (Milliman and



Farnsworth, 2011; Staub and Esterle, 1993). The drainage area of the Rajang Estuary is 6,500 km$^2$,
and 50% is covered with extensive peat in a depth of greater than 3 m (Staub and Gastaldo, 2003).
The climate in the Rajang watershed is classified as tropical ever-wet type (Morley and Flenley, 1987),
while the precipitation varies between dry and wet seasons. Water discharge rate for the Rajang River
reaches 6000 m$^3$ s$^{-1}$ in the wet season (December to March), with an average discharge of about 3600
m$^3$ s$^{-1}$ (Jeeps, 1963; Staub et al., 2000; Staub and Gastaldo, 2003). Sibu city is assumed to be the
boundary between the Rajang drainage basin and the Rajang Estuary (Staub et al., 2000; Staub and
Esterle, 1993). Apart from mineral soils from the upper stream, the Rajang Estuary also receives a
materials from the adjacent hill regions and the Retus River (Staub and Gastaldo, 2003). There are
several tributaries for the Rajang River in the estuary, including Igan, Serendeng, and Rajang. The
Igan tributary is the major outlet for freshwater (Jiang et al., 2019). Mangroves distributed in the
brackish-water area in the southwestern of the estuary. *Casuarina* was observed in the northeastern
and coastal area (Scott, 1985). The thick coverage of vegetation, especially mangroves, in the Rajang
Estuary produces the high-ash, high-sulfur, degraded sapric peats (Lampela et al., 2014). Tide is
diurnal to semidiurnal type in the Rajang Estuary and could extend to Sibu city (Staub et al., 2000;
Staub and Gastaldo, 2003). The range increases from the northeast (1.5 m) to the southwest (2.5 m).
Sediments in the Rajang Estuary are composed of gley soils, podzols soils, and alluvia soils (Staub
and Gastaldo 2003). Gley consists of mixed-layered illite-smectite, illite, and chlorite. Gley is
frequently observed in the central and southwestern part of the estuary (Staub and Gastaldo, 2003).
Podzols is gray-white to white clay, which is composed of kaolinite and illite. Podzols is found in
some low-lying areas and the landward part of the Rajang Estuary (Staub and Gastaldo, 2003).
Alluvial soils, which is made up of illite, smectite, and kaolinite, is found in the landward part of the
estuary (Staub and Gastaldo, 2003). The input of total suspended solids from the Rajang River is up
to 30 Mt yr$^{-1}$ (Milliman and Farnsworth, 2011).
Peatland-draining rivers (Maludam, Simunjan, Sebuyau) are blackwater rivers, characterized by tea-
color, acidic, and oxygen deficit as described by Kselik and Liong (2004). The Maludam River is a
pristine river with minor human influences, since the majority of the river is located in the Maludam
National Park (the second largest park in Sarawak). The peat thickness in the river bed reaches 10 m
(Forest Department, 2014). The catchment of Maludam River is 91.4 km$^2$ and the average discharge





is 4.4±0.6 m$^3$ s$^{-1}$ (Müller et al., 2015). However, other two blackwater rivers are undergoing severe
human activities disturbance, mostly from the plantations of commercial crops like oil palm and sago,
as shown in Fig. 1d (Wetlands International, 2010).
**2.2 Sample collection and process**
The sampling stations are outlined in Fig. 1. The surveys in the Rajang River were conducted in
August 2016 (dry season) and March 2017 (wet season). Each survey lasts 4 to 5 days, covering both
floolding tides and ebbing tides. The samples include fresh river samples, brackish water in different
river tributaries and coastal saline water. In the Rajang watershed, the selection of sampling stations
is dependent on the salinity, anthropogenic activities. In March 2017, the blackwater rivers, as
aforementioned, were included in the filed. During the cruises, surface water samples were collected,
using a pole sampler. The front of the sampler was attached to a 1 L high-density polyethylene bottle
(Nalgene). The length of the pole is 3-4 m to avoid the contamination from the ship. Water samples
were filtered through acid-cleaned 0.4 μm pore size polycarbonate membrane filters (Whatman) into
a polyethylene bottle (Nalgene), then frozen at -20℃, and packed in triple bags. The samples then
thawed at room temperature in the clean laboratory and acidified with ultrapure HCl to pH 1.7 in an
ultra clean lab. All bottles used in the sample collection and storage were prepared in the clean
laboratory, by rinsing with Milli-Q water, immersing in 2% Citranox detergent for 24 h, rewashing
with Milli-Q water for 5-7 times, leaching for 7 days in 10% HCl, rinsing with Milli-Q water 5-7
times again, filling 0.06 mol L$^{-1}$ ultrapure HCl for 2 days at 60℃, and sealing in plastic bags.
**2.3 Sample analyses**
The concentration of dFe was preprocessed using the single batch resin extraction and the isotope
dilution method (Lee et al., 2011). It was quantified on a multi-collector inductively coupled plasma
mass spectrometer in high-resolution mode (Neptune, Thermo). The inlet system contained an Apex
IR desolvator (AEI) with a perfluoroalkoxy microconcentric nebulizer (ESI) at a solution uptake rate
of 50 μL min$^{-1}$. All tubes used for the analyses were acid leached for two days with 10% HCl at 60℃,
rinsed 5 times with Milli-Q water, later filled with 0.06 mol L$^{-1}$ ultrapure HCl in a class 100 flow
bench, and leached for another 2 days at 60℃. The analytical procedural blank and detection limit
(three times the standard deviation of the procedural blank) were both 0.06 nmol L$^{-1}$. The accuracy





of the method was tested by analyzing intercalibration samples including one open ocean SAFe D1
and one estuary water SLEW-3. Measured dFe concentrations for SAFe D1 and SLEW-3 were
$0.66\pm0.05$ nmol $L^{-1}$ and $10.0\pm0.4$ nmol $L^{-1}$ compared to consensus values of $0.70\pm0.03$ nmol $L^{-1}$ and
$10.2\pm1.2$ nmol $L^{-1}$ (Zhang et al., 2015).
During the field investigation, salinity, temperature, pH, and dissolved oxygen (DO) concentrations
were detected *in-situ* with a probe (AP2000, Aquared, U.K.). In the Rajang River, suspended
particulate matters (SPM) samples were collected with pre-combusted 0.7 μm pore size Whatman
GF/F filters, and SPM concentration was calculated by the weight difference of filters before and
after filtration. Dissolved organic carbon (DOC) samples were collected by filtering through 0.2 μm
pore size nylon filters. For the samples collected in August 2016, DOC concentrations were
determined on an Aurora 1030W total organic carbon analyzer. Reproducibility for concentrations
was $\pm0.2$ mg $L^{-1}$. DOC concentrations were measured at the Centre for Coastal Biogeochemistry at
Southern Cross University (Lismore, Australia). For the samples collected in March 2017, DOC
concentrations were determined by the high-temperature catalytic oxidation method with Total
Organic Carbon Analyzer (Shimadzu), and the coefficient of variation was 2% (Wu et al., 2013).
**2.4 The calculation of dFe flux and yield**
To estimate the magnitude of dFe flux from tropical rivers to coastal water, the following equation is
adopted:
$$Q = C \times V \times (1 - RF) \qquad (1)$$
where Q is dFe flux, C is the mean dFe concentration at freshwater endmember (S<1), V is the river
discharge, RF is the removal factor, based on the ratio of the integration area of dFe concentration
versus salinity to that of the theoretical dilution line intercepts (Hopwood et al., 2014). Riverine dFe
yield is the ratio of dFe flux to the drainage area.
**3.   Results**
**3.1 Hydrographic properties in the Rajang and blackwater rivers**
In August 2016 (the dry season), the salinity in the Rajang water samples ranged from 0.0 to 32.0,
which increased from Sibu city to the coastal zone (Table 1). In March 2017 (the wet season), the
salinity varied from 0.0 to 30.1 (Table 1). In Serendeng tributary, some high salinity samples inside



the river mouth in the wet season were found. The concentration of SPM ranged from 24.2 mg L$^{-1}$ to
327.2 mg L$^{-1}$, and decreased from freshwater to seawater, but the highest turbidity water varied among
channels and seasons. In August 2016, the SPM peak was observed near the river mouth in Serendeng
tributary but moved landward in other tributaries (Fig. 2b). In March 2017, the peak of SPM was
located in freshwater in the Rajang tributary. DO in March 2017 (mean: 6.1±0.7 mg L$^{-1}$) was higher
than August 2016 (mean: 3.8±0.6 mg L$^{-1}$), and decreased along the transportation in the Rajang
drainage basin as shown in Fig. 2c. The DO distribution in the Rajang Estuary varied between two
seasons. The high value was found in the west estuary in March 2017 (Fig. 2c, 2h). Water pH in the
Rajang River increased along the salinity gradient with a mean value of 7.1±0.5 (August 2016) and
7.1±0.6 (March 2017), as outlined in Fig. 2d and 2i.
In blackwater rivers, salinity ranged from 0.0 to 23.5 in the Maludam River and 0.0 to 13.6 in the
Sebuyau River. The samples in the Simunjan River are only fresh water. All three blackwater rivers
were frequently anoxic (DO < 2 mg L$^{-1}$). The mixing between river water and ocean water markedly
increased DO. Moreover, pH in these blackwater rivers was relatively low, especially in the Maludam
River (minimum 3.7). DOC increased steadily in fresh water but decreased sharply in estuaries. The
distributions of these properties in blackwater rivers are shown in the Supplement.
**3.2 dFe in Rajang and its Estuary**
The contour of dFe in the Rajang surface water is shown in Fig. 2. The dFe concentrations in the
Rajang freshwater ranged from 2.8 to 7.3 μmol L$^{-1}$ (mean: 5.2±1.8 μmol L$^{-1}$) in August 2016, and
ranged from 4.2 to 8.3 μmol L$^{-1}$ (mean: 6.4±2.0 μmol L$^{-1}$) in March 2017. In the Rajang Estuary, the
concentration of dFe ranged from 1.7 nmol L$^{-1}$ to 7.0 μmol L$^{-1}$ (mean: 1.1±2.2 μmol L$^{-1}$) and 4.2 nmol
L$^{-1}$ to 11.3 μmol L$^{-1}$ (mean: 4.2±4.0 μmol L$^{-1}$) in the dry and the wet season, respectively. The
concentration of dFe in the wet season was higher than the dry season both in the Rajang freshwater
and the Rajang Estuary.
The relationships between dFe concentrations and other factors, such as salinity, SPM, DOC, DO and
pH in the Rajang Estuary can be found in Fig. 3. In the dry season, dFe concentration decreased
exponentially in low salinity water (salinity<15) though we did not include the tidal influence. A
linear relationship was found between dFe and SPM in low salinity area (R$^2$=0.29, $p$<0.05). In the
high salinity area (S>15), dFe tended to be conservative (Fig. 3a), and displayed a linear relationship





with DOC ($R^2$=0.45, $p$<0.05), DO ($R^2$=0.50, $p$<0.05), and pH ($R^2$=0.39, $p$<0.05). In the wet season,
dFe concentration was higher in the Igan tributary compared to other two branches. There was an
intensive dFe addition between salinity 5-15, mainly in the Serendeng tributary (Fig. 3a). Specifically,
the linear correlation between dFe and SPM was found in the water samples when salinity was <15
in the wet season ($R^2$=0.11, $p$<0.05) (Fig. 3b), especially in the Serendeng distributary. Moreover, a
significant positive relationship between dFe and DOC was also identified in the wet season in low
salinity water ($R^2$=0.61, $p$<0.001) (Fig. 3c). DO was negatively correlated with dFe in high salinity
area ($R^2$=0.97, $p$<0.001), with a similar pattern in the dry season. The relationship between pH and
dFe was insignificant in the wet season.
**3.3 dFe in blackwater rivers**
The average dFe concentration in three blackwater rivers were 14.6±6.7 μmol L$^{-1}$ (the Maludam
River), 44.2±11.8 μmol L$^{-1}$ (the Simunjan River), and 17.6±12.0 μmol L$^{-1}$ (the Sebuyau River). The
dFe concentration increased along the river flow (Fig. 4a), but decreased during the mixing. The
distribution of dFe in blackwater rivers tended to be conservative in the estuary of Maludam and
Sebuyau (Fig. 4b), which was different from the pattern in the Rajang Estuary. Moreover, there was
a significant positive correlation between dFe and DOC in the blackwater rivers (Fig. 4c), except the
Maludam River because of low DOC in high salinity region (S=20.0).
**4. Discussion**
**4.1 Seasonal variation of dFe in the Rajang freshwater**
In the dry season, dFe concentrations in the Rajang water (near the Sibu city) ranged from 2.8 to 7.3
μmol L$^{-1}$. In the wet season, dFe concentrations increased (Fig. 2). Considering the limited
temperature variation in the tropical zone, the dFe elevation may be related to the stronger weathering
derived from intensive precipitations. In particular, once precipitations elevated (in the wet season),
water mass from the upper stream scoured the soils along the river banks, carring the Fe-enriched
terrestrial particles to the down stream (Meade et al., 1985; Taillefert et al., 2000). Moreover, the
agriculture activities in the watershed, such as tillage, can result in the rapid leaching in the wet season
(Lehmann and Schroth, 2003; Tabachow et al., 2001), especially in 2017 (the occurance of La Niná
events) (Jiang et al. 2019). Additionaly metal elements were transported from the catchement to the





Rajang River (Johnes and Hodgkinson, 1998; Withers et al., 2001). In addition, the changes of soil
structure during agriculture activities can influence the exchange route of dissolved matters in vertical
profiles; hence the large proportion of dFe is likely to be transported during the rainfull via water
exchanges (Haygarth et al., 1998; Johnes and Hodgkinson, 1998). Such dFe addition from the
cropland was also observed in many other study areas, like the Krishna river drainage area (Kannan,
1984), the Palar and Cheyyar river basin (Rajmohan and Elango, 2005), and the Guadalquivir River
(Lorite-Herrera and Jiménez-Espinosa, 2008). Eventually, the terrestrial -borne dFe injected into the
Rajang River via hydrological connections in the riparian ditches, and hence contributed quantities
of dFe to rivers from terrestrial runoff and flood discharges (Yan et al., 2016).
**4.2 dFe in the Rajang Estuary**
In the Rajang Estuary, there was an intensive removal of dFe when the salinity < 15, especially in the
dry season (Fig. 3a). This may be mainly related to the flocculation of the negatively charged colloids
with cations in the fresh-saline water mixing. This has been observed in many rivers and simulation
experiments (Boyle et al., 1977; Oldham et al., 2017; Zhu et al., 2018). Furthermore, dFe was
negatively correlated with SPM in low salinity waters (Fig. 3b), indicating that the dFe removal may
also be linked to the absorption of SPM as described by other researches (Beusekom and Jonge, 1994;
Homoky et al., 2012; Zhang et al., 1995). However, there was exceptionally high dFe concentration
at salinity 5-15 in the Serendeng tributary in the wet season. On the one hand, it may result from
peatland soils in the adjacent area, because the peatland soils host abundant dFe and organic ligand,
and these organic compounds could enhance the solubility of Fe during the transport (Krachler et al.,
2010; Oldham et al., 2017; Shuhaimiothman, 2009). On the other hand, there could be other processes
for dFe addition in the Rajang Estuary, such as the desorption of SPM-bounded Fe to the river water.
The balance between adsorption and desorption of trace metal ions onto/from SPM is complicated.
These two processes could occur simultaneously and be influenced by different environmental
conditions, like SPM content, pH, salinity, and adsorption strength between ions and SPM (Hatje et
al., 2003; Jiann et al., 2013; Zhang et al., 2008). It has been confirmed that the partition coefficient
of dFe decreased with increasing SPM concentration, and became inversely proportional to the SPM
concentration, termed as particle concentration effect (Benoit, 1995; Jiann et al., 2013; Turner and
Millward, 2002). Furthermore, Zhu et al. (2018) suggested that desorption from particles was the



main reason for dFe enhancement in the river mouth of the Changjiang. Although we didn't conduct
any adsorption/desorption experiments in these cruises, we could deduce that the great dFe increase
at salinity 5-15 in the wet season may be also related to the desorption because of the high SPM
content. Moreover, the samples in these area was collected during a spring tide, and the high
concentration of SPM was within the range of the result in the Texas river (Jiann et al., 2013) and
some observations in the Changjiang estuary (Zhu et al., 2018), where dFe enhancements resulted
from the desorption of SPM. Furthermore, the intensive agricultural activities in the Serendeng
distributary changed the soil structure and contributed a considerable amount of SPM at flood tide,
which may stimulate the dFe desorption.
In the high salinity zone (S>15), the dFe tended to be conservative. The positive relationship between
dFe and DOC in the dry season (Fig. 3c) may be a mirror of the chemical association of dFe and
DOM. Specifically, the combination betwen dFe and organic matter, especially the pelagic organic
matter, can resist to salt-induced aggregation and lead to an input of bioavailable dFe to the coastal
zone (Breitbarth et al., 2009; Krachler et al., 2005; Stolpe and Hassellov, 2007).
The multiple linear regression analysis of dFe and environmental factors, including salinity, SPM,
DOC, DO, and pH (dry season: $R^2$=0.52, $p$<0.05; wet season: $R^2$=0.73, $p$<0.05), was also used to
explore the observed patterns. It showed that salinity and SPM were the main factors for the
distribution of dFe in the Rajang Estuary ($p$<0.05). For pH, the correlation between dFe and pH was
limited in the wet season, suggesting a little impact of pH on dFe. In the dry season, the concentration
of dFe was negatively correlated with pH (Fig. 3e), becasue Fe-enriched sediments can be acidized
and mineralized by inorganic acids ($H_2CO_3$, $HNO_3$, and $H_2SO_3$) and organic acids (oxalic acid, citric
acid, and siderophore) generated from the chemical weathering and biological progress (Banfield et
al., 1999; Lerman et al., 2007). The biogeochemical behavior of dFe in the Rajang River that we
discussed above is summarized and conceptualized in Fig. 6a.
**4.3 dFe in blackwater rivers**
In blackwater rivers, the dFe was accumulated from the upper stream to the downstream before
mixing. In the mixing zone, high concentrations of dFe were rapidly diluted (Fig. 4b). As evidenced
by the water color, these peat-draining rivers are characterized by extremely high levels of terrigenous
DOM (Martin et al., 2018; Zhou et al., 2018).Given such high concentrations of DOM and the positive





correlation between dFe and DOC (Fig. 4c), peatland should be a strong source for dFe. Consequently,
the gradual enrichment of dFe along the rivers was observed. Compared with the Maludam River, i.e.
the drainage from an undisturbed peatland, dFe concentrations in the Sebuyau River and the Simunjan
River were significantly higher (Table 1). The difference in the dFe concentration among three
blackwater rivers may come from the variation of environmental parameters around the drainage
basin, especially the vegetation types and anthropogenic activities. The palm oil plantations covered
a significant area in the watershed of the Sebuyau River and Simunjan River, as shown in Fig. 1d. In
order to stimulate seedings in plantations, the empty fruit bunches and the palm oil mill effluent were
returned to the cropland after oil extraction (Carron et al., 2015; Nelson et al., 2015), indicating an
enhancement in terrestrial organic matter. The intensive agriculture activities, such as tillage, also
facilitated the transport of terrestrial dFe into the Sebuyau River and the Simunjan River as discussed
in chapter 4.1.
During the cruise, the high salinity samples were not obtained in the Maludam and Sebuyau rivers.
For the samples with the salinity range from 0 to 20, the dFe removal is insignificant, which is
markedly different from the trend obtained in the Rajang Estuary (Fig. 4b). The significant positive
correlation between dFe and DOC concentration reinforced the tight connection between dFe and
organic ligands in blackwater rivers (Fig. 4b). Recent studies have also pointed out that organic
ligands originating from peatland enhanced the iron-carrying capacity of the river water (Krachler et
al., 2005; Oldham et al., 2017). Approximately 20% of dFe didn't flocculate during a laboratory
mixing experiment (Krachler et al., 2010). The biogeochemical behavior of dFe in blackwater rivers
that we discussed above is summarized and conceptualized in Fig. 6b.

**4.4 dFe fluxs and yields**

For the Rajang River, the mean dFe concentration at the river endmember of two seasons was 5.5±2.0
μmol L$^{-1}$, and mean removal factor was 98.0±0.6%. Removal factor of dFe varied on a global scale.
The Rajang RF was predominant among the recent results (Table 2). Coupled with the discharge rate
(about 3600 m$^3$ s$^{-1}$), the dFe flux from the Rajang River was estimated to be (6.4±2.3)×10$^5$ kg yr$^{-1}$
besed on the equation (1). For the Maludam River, the concentration of river endmember was
14.6±6.8 μmol L$^{-1}$, and RF=0 due to the conservative distribution. The dFe flux in the Maludam River
was approximately (1.1±0.5)×10$^5$ kg yr$^{-1}$, produced from 432 km$^2$ peatland in the Maludam National



Park. It is the same magnitude with the Rajang dFe flux, suggesting that the dFe input were
considerable in blackwater rivers. Malaysia hosts peatland area about 25,889 km$^2$, the dFe flux can
be $(6.6\pm3.0)\times10^6$ kg yr$^{-1}$ on the basis of the yield from the Maludam River. Consequently, the
blackwater rivers contributed 10 times greater dFe than the Rajang River to the coastal zone in
Malaysia, even though their discharges are small (Milliman and Farnsworth, 2011). This terrestrial
dFe may play an important role in supporting primary producers in the adjacent ocean (Breitbarth et
al., 2009; Laglera and Berg, 2009).
The concentration and yield of dFe varied among tropical rivers as shown in Fig. 5. Compared with
subtropical rivers, like the Changjiang (Zhu et al., 2018) and the Mississippi River (Shiller, 1997;
Stolpe et al., 2010), the tropical rivers contributed a great amount of dFe, such as the Amazon River
(Aucour et al., 2003; Bergquist and Boyle, 2006) and the Congo River (Coynel et al., 2005; Dupré et
al., 1996). For rivers hold a similar discharge rate and drainage area with the Rajang River, like the
Fraser River, a temperate river in Canada, dFe concentrations and yield was significantly lower than
that derived from the Rajang River (Cameron et al., 1995). One reason for the high concentration and
yield in tropical rivers likely results from intensive weathering, leaching of the rocks and sediments,
and abundant plantations under high temperature and heavy precipitations (Bergquist and Boyle, 2006;
Fantle and Depaolo, 2004). Compared with other tropical rivers, such as the Amazon River and the
Congo River, the dFe concentration in the Rajang River was similar, however dFe yield was lower in
the Rajang River. This may be related to the difference of plantation types (Aucour et al., 2003;
Coynel et al., 2005; Dupré et al., 1996). The peatland soils in the Rajang Estuary may contribute to
the higher dFe yield, as the Niger River passing through a dry savanna (Picouet et al., 2002). Different
from the Niger River, the Senaga River drains from a savannah-rainforest area, and contains a
considerable amount of SPM, similar to the Rajang River. The dFe yield was comparable with the
Rajang River. As for some small tropical rivers, like the Swarna River (Tripti et al., 2013), the Nyong
River (Olivié-Lauquet et al., 1999), the Periyar River (Maya et al., 2007) and the Chalakudy River
(Maya et al., 2007), the dFe concentration was similar to the Rajang River, but with higher dFe yields
and DOC concentrations. In these small tropical rivers, the drainage basins were covered with organic
matter enriched sediments, which may be a great source of dFe.
In blackwater rivers, the dFe concentration and yield were much higher than the records from Rajang



River. The high concentration of DOM is likely to be the main reason of high dFe contents in the
blackwater rivers, such as the situations in the Kiiminkijo River (Heikkinen, 1990), the Tannermoor
River (Krachler et al., 2005), the Halladale River (Krachler et al., 2010), the Bebar River (Gastaldo,
2010), and the Taieri River (Hunter, 1983) (Fig. 5b). The human impacts, such as agricultural
activities and the plantations of oil palm, may also contribute to a bulk of dFe to the blackwater rivers.

**5. Conclusions**
In this study, dFe was investigated in the Rajang River and three blackwater rivers in Sarawak,
Malaysia. The conclusions are as follow:
1. There was a significant seasonal variation of dFe concentration in the Rajang freshwater with a
higher dFe concentration in the wet season, likely due to the increased leaching and terrestrial
erosion. The dFe removal was intensive in low salinity area (salinity<15) of the Rajang Estuary
due to the salt-induced flocculation and absorption onto the SPM. On the contrary, dFe tended to
be conservative in the high salinity area (salinity>15), which may be due to the binding between
dFe and the organic matter. In addition, there were significant additions of dFe in some tributaries
due to the desorption of SPM and anthropogenic inputs.
2. The concentration of dFe in the blackwater rivers was 3-10 times higher than that of the Rajang
River, which was related to the contribution of peatland soil. Anthropogenic activities in the
watershed also influenced the dFe concentration in blackwater rivers. Different from the pattern
observed in the Rajang River, there wasn't remarkable dFe removal in blackwater river estuaries.
3. The dFe yield in blackwater rivers was much higher than that of the Rajang River. This result
indicated that the dFe flux from blackwater rivers can be crucial for coastal zones in Malaysia.
This study improved the understanding of dFe distribution in the Rajang River and confirmed its
regional influence. In addition, we provided the direct evidence that blackwater rivers had an
extremely high yield of dFe. Furthermore, anthropogenic activities may have a critical impact on the
concentration and distribution of dFe in these tropical rivers in Malaysia.

**Acknowledgment**
The present study was kindly funded by the National Natural Science Foundation of China
(41476065). Further funding was provided under the MOHE FRGS 15 Grant



(FRGS/1/2015/WAB08/SWIN/02/1), SKLEC Open Research Fund (SKLEC-KF201610) and
Overseas Expertise Introduction Project for Discipline Innovation (111 Project, B08022). We would
like to thank the Sarawak Forestry Department and Sarawak Biodiversity Centre for permission to
conduct collaborative research in Sarawak waters under permit numbers NPW.907.4.4 (Jld.14)-161,
Park Permit No WL83/2017, and SBC-RA-0097-MM. Thanks to Lukas Chin and the "SeaWonder"
crew for their support during the cruises. Technical support by Dr. Patrick Martin and Dr. Gonzalo
Carrasco at Nanyang Technological University during the cruises and Ms. Yun Xue, Ms. Shuo Jiang
and Ms. Wanwan Cao at East China Normal University in the laboratory analysis are also gratefully
acknowledged.

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

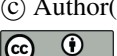



Table 1. Range and average of Salinity (S), pH, suspended particulate matter (SPM), dissolved oxygen (DO), dissolved iron (dFe), and dissolved
organic carbon (DOC).

| River-Time | Station | S | pH | SPM (mg L$^{-1}$) | DO (mg L$^{-1}$) | dFe (µmol L$^{-1}$) | DOC (µmol L$^{-1}$) |
|---|---|---|---|---|---|---|---|
| Rajang-August 2016 | 28 | 0-32.0 | 6.5-8.1 | 24.2-129 | 2.7-4.8 | 0.002-7.3 | 181-357 |
| (dry season) | | (11.7±12.1) | (7.2±0.5) | (62.3±30.4) | (3.8±0.6) | (2.3±2.9) | (218±78.2) |
| Rajang-March 2017 | 15 | 0-30.1 | 6.0-7.1 | 47.1-327 | 4.6-7.6 | 0.004-11.3 | 98.1-238 |
| (wet season) | | (11.9±12.3) | (7.1±0.7) | (151±70.4) | (6.1±0.7) | (4.6±4.1) | (165 ± 41.8) |
| Maludam-March 2017 | 9 | 0-20.0 | 3.7-7.6 | 0.4-388 | 1.1-6.8 | 6.3-23.8 | 353-4581 |
| | | (5.4±6.1) | (4.6±1.4) | (53.1±121) | (2.7±1.9) | (14.6±6.8) | (3609±1229) |
| Sebuyau-March 2017 | 8 | 0-13.6 | 4.3-7.0 | 0.4-388 | 1.4-5.9 | 3.0-33.6 | 364-2078 |
| | | (5.4±6.1) | (5.2±1.1) | (53.1±121) | (3.2±1.9) | (17.6±12.0) | (1396±671) |
| Simunjan-March 2017 | 6 | 0-0.4 | 4.7-6.3 | 14-481 | 1.0-2.6 | 25.8-59.2 | 818-3121 |
| | | | (5.2±0.6) | (135±197) | (1.9±0.7) | (44.2±11.8) | (2157±950) |






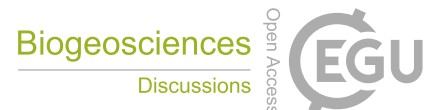

Table 2: Concentration of dFe and removal factor (RF) in some rivers.

| Rivers | Estury location | Climate | dFe (μmol/L, '*' in nmol/L) | RF (%) | Reference |
|---|---|---|---|---|---|
| Lena | Russia | arctic | 0.54 | 67.5 | 1, 2, 3 |
| Changjiang | China | subtropical | 44.6* | 79.1 | 1, 4 |
| Jiulongjiang | China | subtropical | 17.9* | 37.7 | 5 |
| Columbia | United States | subtropical | 71.4* | 72.5 | 6 |
| Garonne | France | temperate | 0.1 | 59.7 | 7 |
| Merrimack | United States | temperate | 3.7 | 44.6 | 1.8 |
| Amazon | Brazil | tropical | 1.9 | 77.8 | 1.9.10 |
| Congo | Congo | tropical | 3.2 | 57.3 | 1,11,12 |
| Rajang | Malaysia | tropical | 5.5 | 98 | 1, this study |

1. Milliman and Farnsworth, 2011; 2. Martin et al., 1993; 3. Guieu et al., 1996; 4. Zhu et al., 2018; 5. Zhang 1995; 6. Bruland et al., 2008; 7. Lemaire et al., 2006; 8. Boyle et al., 1974; 9. Aucour et al., 2003; 10. .Moreira-Turcq et al., 2003; 11. Dupré et al., 1996; 12. Coynel et al., 2005.

* RF is the ratio of the integration of dFe concentration versus salinity and the product of theoretical dilution line intercepts (Hopwood et al., 2014).

* dFe yield is a ratio of dFe flux and drainage area.





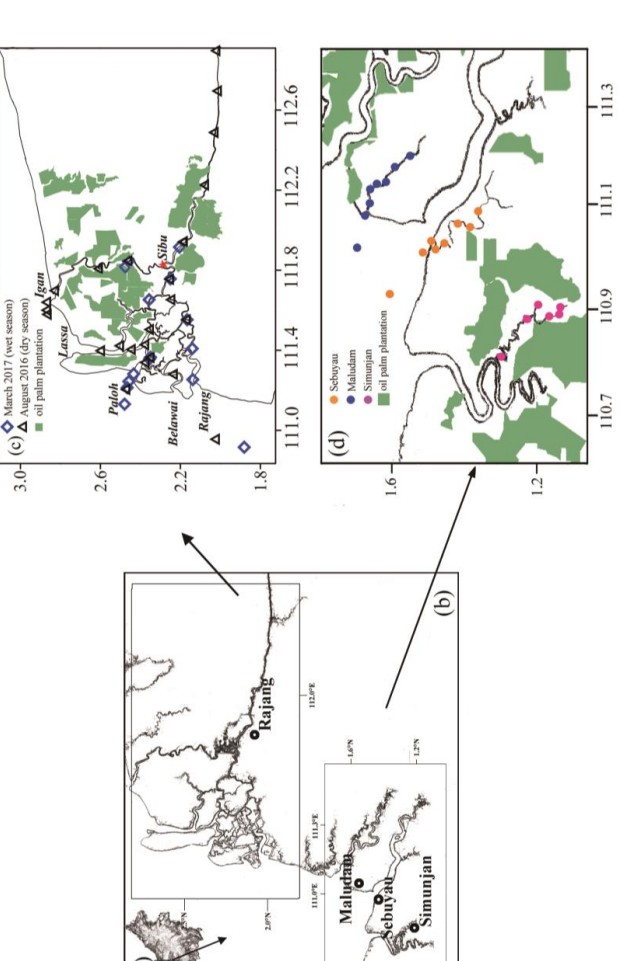

Figure 1. Locations of sample stations in Malaysia (a), including the Rajang River, the Maludam River, the Sebuyau River, and the Simunjan River (b). In figure (c) and (d), the green layer indicates oil palm plantations, based on the dataset from Global Forest Watch (http://gfw2-data.s3.amazonaws.com/country/mys/zip/mys_oil_palm.zip).

**Biogeosciences** Open Access
Discussions
EGU

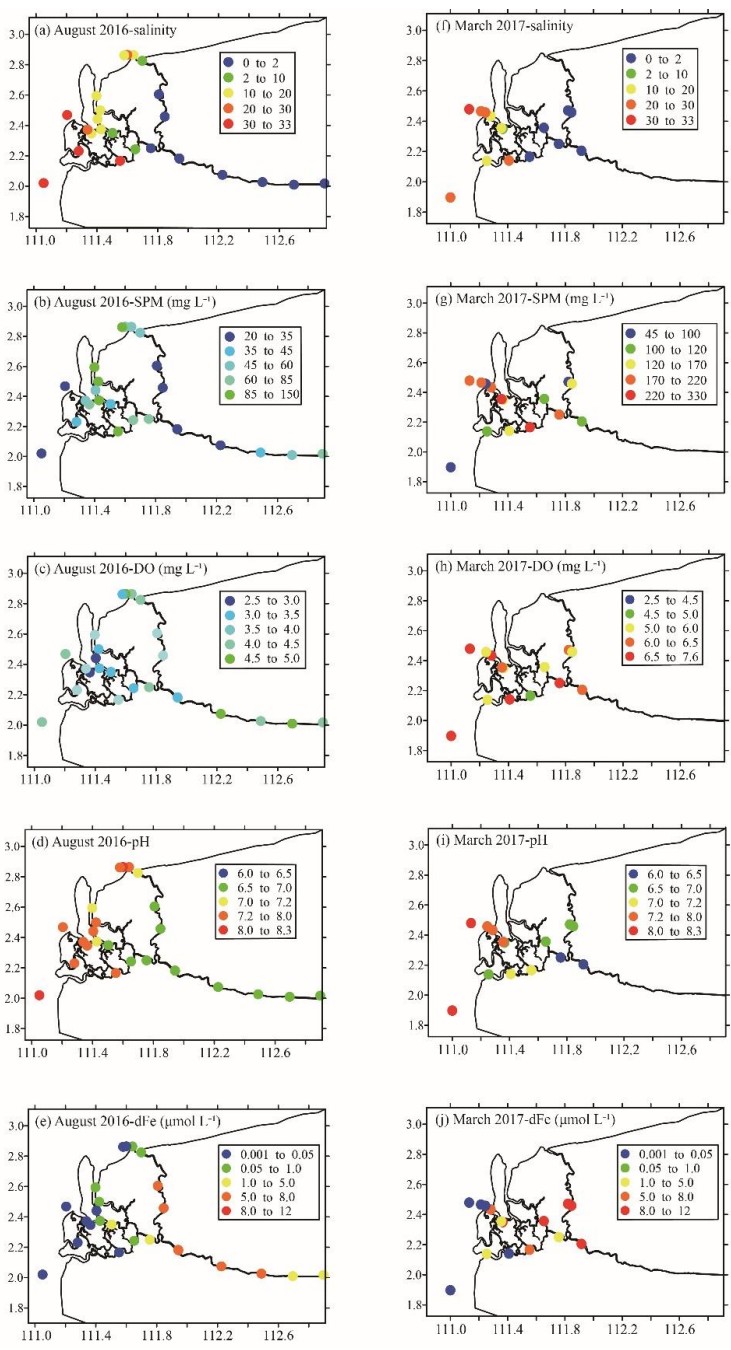

Figure 2. Spatial distributions of salinity (a) (f), suspended particulate matter (SPM) (b) (g), dissolved oxygen (DO) (c) (h), pH (d) (i), dissolved iron (dFe) (e) (j) in the Rajang River in August 2016 and March 2017.

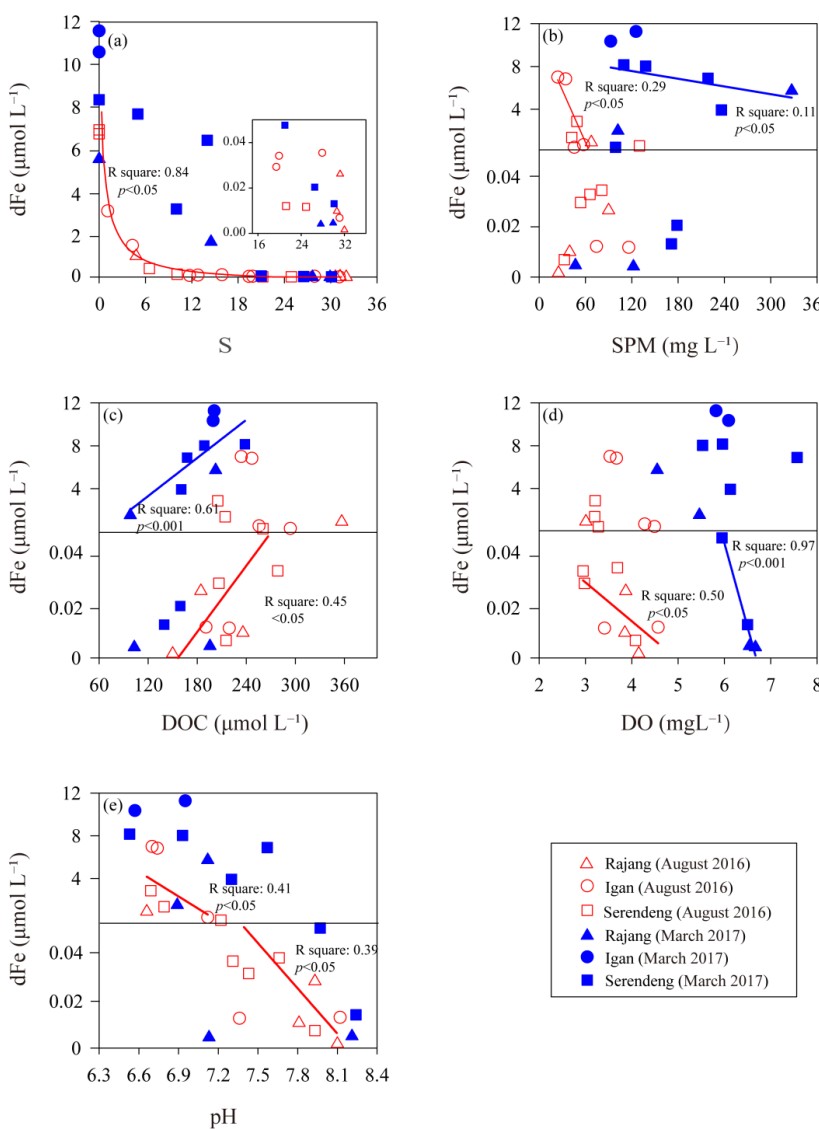

Figure 3. Dissolved iron (dFe) correlation with salinity (S) (a), suspended particulate matter (SPM) (b),
dissolved organic carbon (DOC) (c), dissolved oxygen (DO) (d), and pH (e) in Rajang estuary. The solid
lines were the linear regressions between dFe and other factors, and the colors of the lines were coincident
with the data points in different salinity range.


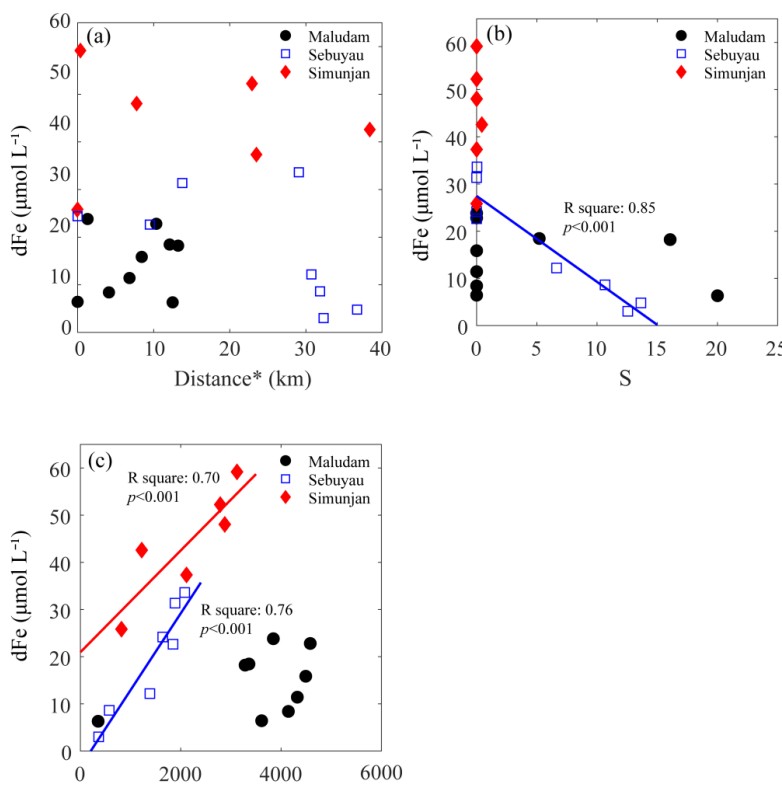

Figure 4. the correlations between distance (a), salinity (S) (b), dissolved organic carbon (DOC) (c), and
dissolved iron (dFe) in blackwater rivers: Maludam, Sebuyau and Simunjan. The solid lines were the linear
regressions between dFe and other factors, and the colors of the regression lines were coincident with the
data points.

*We adopted the station at the upper stream as distance=0, and the downstream direction as positive.





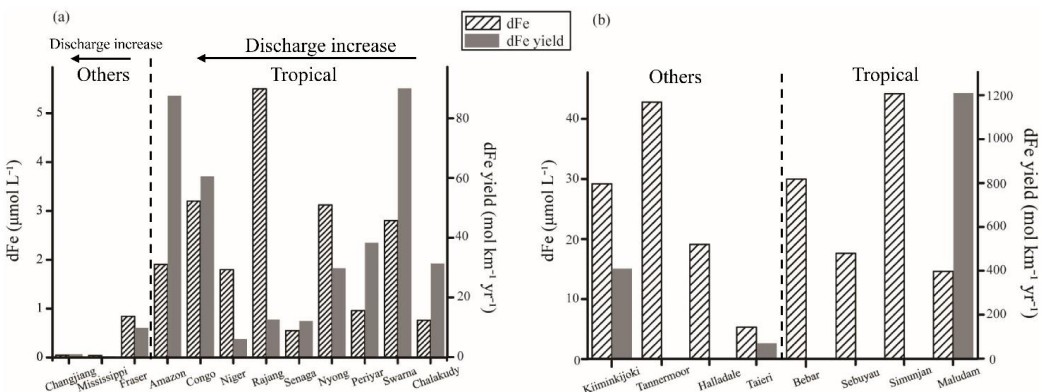

Figure 5. The concentration and yield of dFe in large rivers (a) and blackwater rivers (b).




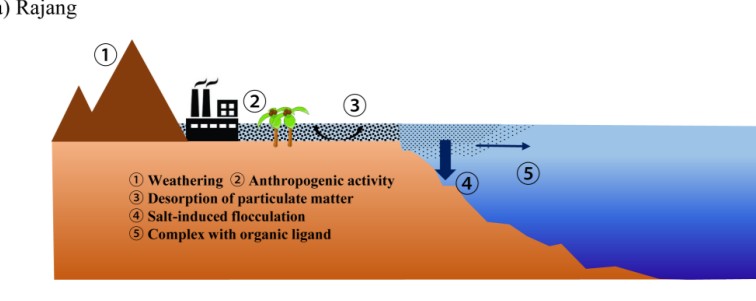

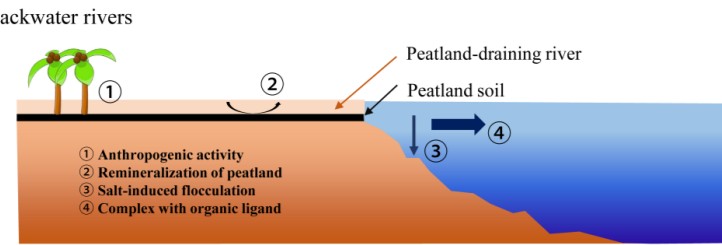



Figure 6. A schematic representation of dFe biogeochemical behaviors in the Rajang River (a) and
blackwater rivers (b)