# Peer review of "Distribution and Flux of Dissolved Iron in the Peatland-draining Rivers"

_Biogeosciences, 2019_

## Referee Comment (RC1) · Anonymous Referee #1 · 7 Sep 2019

This article has discussed the distribution and flux of dissolved iron of the Rajang and Blackwater Rivers at Sarawak, Borneo. The idea of the article is clear and the analysis is thorough. The following points should be issued: 1. In Section 2.3, two methods for determination of DOC were mentioned. Though these two methods are all acceptable, but I think it is better to use the same method in one work for comparison. 2. Did the authors consider the effect of temperature on dFe? 3. In Section 4.1, there were not only seasonal variation analysis but also spatial analysis. In order to summary the corresponding analysis content more accurate and keep pace with the following several sections, I think it is better to entitle Section 4.1 as "dFe in the Rajang freshwater". 4. In Page 8, Line 201-202,"In the Rajang Estuary, the concentration

of dFe ranged from 1.7 nmol L-1 to 7.0 $\mu$mol L-1 (mean: 1.1 $\pm$ 2.2 $\mu$mol L-1)", Is it correct? 5. As shown in Table 1, the standard deviations for SPM and DOC are rather large. Is it due to the different samples? 6. One or more relative references about "the stronger weathering derived from intensive precipitations" should be added to make the statement of "Considering the limited temperature variation in the tropical zone, the dFe elevation may be related to the stronger weathering derived from intensive precipitations. (Page 9, Line 230-232)" more convincing (needing literature support). 7. Check the manuscript carefully. There are several language errors in this paper, for example "supporing" in Page 3 Line 42 and "carring" in Page 9 Line 233. CO2(In Page 21,Line 569) should be changed to CO2.

Please also note the supplement to this comment:
https://www.biogeosciences-discuss.net/bg-2019-204/bg-2019-204-RC1-supplement.pdf

---

## Referee Comment (RC2) · Anonymous Referee #2 · 17 Sep 2019

The authors present dissolved iron (dFe) in the Rajang and the other three blackwater rivers and estuaries in Malaysia. Few results are available for the behavior of dFe in the tropical rivers in Southeast Asia, although those rivers account for large proportions of fluvial discharge and are significant source of terrestrial materials to the oceans. Authors present the precious and accurate data of dFe in the highly dynamic peat-draining Rajang and the other three blackwater rivers and estuaries for the comprehensive understanding of dFe biogeochemical cycle in the tropical regions. As such, their work is citable and appropriate for publication in Biogeosciences. However, there are some unclear information and improper discussions in the manuscript, and moderate scientific revisions, required.

[Figure]

Detailed comments are listed as follow: 1. Based on the description in the 2.1 section, study area covers the drainage basin and estuaries of Rajang River, with major stations locating in the estuaries. So the title should include the "estuaries" inside. 2. The description of the study area and Figure 1 are not clearly enough to help readers to understand the complex riverine drainage and estuaries. For example: a) Authors should mark the location of Sarawak State in Figure 1a or 1b; b) Normally we don't use bold circle to present riverine drainage basin, it looks like the capital city. Authors should give enough information of different mainstreams and tributaries of discussed riverine basin in Figure 1 to help readers to understand the results and discussion later; c) In line106 (page 5), authors mention that "The Igan tributary is the main outlet for freshwater (of Rajange River)". However, the survey carried out in the wet season (March 2017) only had one station in this tributary. There was only one sampling station in the main stream of Rajang River in wet season too. Authors should explain the sampling strategy between two voyages. Otherwise the discussion to compare the results between two seasons are hard to understand due to different stations' coverage. d) In Figure 3, authors don't explain how to separate the stations in the Rajang Riverine basin into three parts of "Rajang, Igan and Serendeng"? And where is "Serendeng" in Figure 1? How do authors separate riverine stations and estuary stations between two sampling voyages? According to the variations of salinity or using Sibu city as the boundary (Page 5 Line 101-102)? Relevant information is very important for the understanding of results of different average concentrations and behavior of dFe between dry and wet seasons. e) Since the behavior of dFe is strongly affected by the SPM based on the discussion, authors should give the information of sediment loads of Rajang River and the other three blackwater rivers. f) What's the water depth of the Rajang River? Since only surface samples were collected and authors explain the difference of dFe behavior between two seasons based on the resuspension and adsorption/desorption on the surface of particles. It's better to give the basic information of Rajang River. 3. Authors prove the robust of dFe measurement by the international intercalibration experiments results using SAFe D1 and SLEW-3. dFe measurements

using isotope dilution ICP-MS method have good precision and accuracy. However, the samples collected in the Rajang River and the other three blackwater rivers have high content of DOM which is quite different with the open ocean and coastal region. What's species of dFe can be measured using solid extraction method depends on the selection of resin and the composition of organic ligands in the riverine and estuarine samples. Authors should give detailed information of resin type in section 2.3 and give clear definition of what's the speciation of measured "dFe" and what's the difference between measured "dFe" and total dissolved Fe (measured after digestion) in the section of results and/or discussion. 4. Page 8 Line 182-190: Authors discussed the distributions of SPM, DO and pH in the two seasons of the Rajang River and Estuary. However, it's hard to follow because we don't know locations of Serendeng tributary and Rajang tributary. The difference of color bar in Figure 2b and 2c is hard to recognize. 5. Section 3.2: As mentioned in the comments 2-d, I don't know how do authors separate the riverine and estuarine stations between dry and wet seasons, the data of concentration ranges and averages are hard to understand due to different sample coverage and data number. Ordinate of the iron concentration in figure 3 using two different scales (0.05?-12 ïA■M and 0-0.05? ïA▯M), what's the scientific hypothesis of this separation? Using salinity >15 and <15 as the separation boundary? If this is true, authors should differentiate the salinity using the same boundary in Figure 2. 6. Section 4.1 page 9 line 231-232: I can't agree with the statement of "dFe elevation may be related to the stronger weathering derived from intensive precipitation". Weathering index in the same region between dry and wet season can't change significantly. The elevation of dFe in the wet season might be caused by the dissolution of weathering product in the riverine drainage basin due to the runoff variation and precipitation, and might also due to the resuspension of bottom sediments. 7. Authors using correlations of dFe with other parameters (e.g. SPM, DO etc) to discuss the behavior of dFe in the riverine and estuary region and between two seasons. Discussion is relatively superficial and need to add some solid evidence.

---

## Author Comment (AC1) · 22 Oct 2019

**Responses to Referees**

**Anonymous Referee #1**

*1. In Section 2.3, two methods for the determination of DOC were mentioned. Though these two methods are all acceptable, but I think it is better to use the same method in one work for comparison.*

**Reply:** thank you for your advice. The samples from two cruises were collected and stored in the same way, but determined in two methods in two research groups. This could be inappropriate and we will keep that in mind in our future research. In August 2016, DOC concentrations were determined via an Aurora 1030W total organic carbon analyzer. Glucose of known concentrations in Milli-Q was used as a standard to correct for drift and to verify sample concentrations, and the reproducibility for concentrations was ±0.2 mg L$^{-1}$. In March 2017, DOC concentrations were determined by the high-temperature catalytic oxidation method with Total Organic Carbon Analyzer (Shimadzu). Furthermore, Deep Sargasso Seawater CRM with a DOC concentration of 41-44 μmol L$^{-1}$ carbon was tested before each batch of samples, and good agreement was obtained between the certified values and the measured values. The standard calibration indicated that both methods can get reliable DOC concentration results, and have published results in several papers.

**Revised manuscript:** Page 7, line 181 to 188: For the samples collected in August 2016, DOC concentrations were determined via an Aurora 1030W total organic carbon analyzer at the Centre for Coastal Biogeochemistry in Southern Cross University (Lismore, Australia). Reproducibility for concentrations was ±0.2 mg L$^{-1}$. For the samples collected in March 2017, DOC concentrations were determined by the high-temperature catalytic oxidation method with Total Organic Carbon Analyzer (Shimadzu) at the State Key Laboratory of Estuarine and Coastal Research in East China Normal University (Shanghai, China), and the coefficient of variation was 2% (Wu et al., 2013).

*2. Did the authors consider the effect of temperature on dFe?*

**Reply:** thank you for your advice. The climate of Sarawak is classified as the tropical ever-wet type with a minor temperature variation (section 2.1). The water temperatures of dry season and wet season were 29.9±1.4°C and 28.1±1.3°C, respectively. This variation may influence the biogeochemical reactions of dFe, but it was negligible compared with other environmental factors, like precipitations. However, we can't ignore the temperature effect of dFe in other research areas, especially poles zone, where the temperature will stimulate the thaw of ice and snow, which is a great source of dFe in polar regions (Klunder et al., 2012).

*3. In Section 4.1, there was not only seasonal variation analysis but also spatial analysis. In order to summary the corresponding analysis content more accurate and keep pace with the following several sections, I think it is better to entitle Section 4.1 as "dFe in the Rajang freshwater".*

**Reply:** thank you for your advice. We have changed. And Section 4.2 was entitled as well.

**Revised manuscript:** Page 10, line 250: 4.1 Seasonal and spatial variation of dFe in the Rajang River. Page 11, line 270: 4.2 Seasonal and spatial variation of dFe in the Rajang Estuary.

*3. In Page 8, Line 201-202,"In the Rajang Estuary, the concentration of dFe ranged from 1.7 nmol L$^{-1}$ to 7.0 µmol L$^{-1}$ (mean: 1.1±2.2 µmol L$^{-1}$)", Is it correct?*

**Reply:** Thank you. The average concentration of dFe in the Rajang River is 5.5±1.7 µmol L$^{-1}$ in the dry season, and the concentration of dFe in seawater endmember (S>30) was 2-10 nmol L$^{-1}$ near the Rajang Estuary. There was an intensive   removal of dFe mainly due to the flocculation of the negatively charged colloids with cations in the mixing process, and the absorption of SPM in the Rajang Estuary as discussed in Section 4.2. The removal factor of Rajang was 98.0%, which was predominant among the recent results. In order to reduce the misunderstandings of the concentration range and mean value of the riverine and estuary, we separated the stations into the Rajang River and the Rajang Estuary (Sibu as the boundary) in Table 1, and make the results consistent between the Table 1 and Section 3.2.

**Revised manuscript:**

Page 9, Line 219 to Line 219. The dFe concentrations in the Rajang River ranged from 3.3 to 7.3 µmol L$^{-1}$ (mean: 5.5±1.7 µmol L$^{-1}$) in August 2016, and ranged from 4.2 to 8.3 µmol L$^{-1}$ (mean: 6.4±2.9 µmol L$^{-1}$) in March 2017.

Page 26, Table 1:

Table 1. Range and average of Salinity (S), pH, suspended particulate matter (SPM), dissolved oxygen (DO), dissolved iron (dFe), and dissolved organic carbon (DOC).

| River-Time | Station | S | pH | SPM (mg L$^{-1}$) | DO (mg L$^{-1}$) | dFe (µmol L$^{-1}$) | DOC (µmol L$^{-1}$) *in mmol L$^{-1}$ |
|---|---|---|---|---|---|---|---|
| Rajang River-August, 2016 | 8 | 0 | 6.7-6.8 (6.7±0.05) | 31.4-95.2 (51.5±22.1) | 3.4-4.8 (4.4±0.4) | 3.3-7.3 (5.5±1.7) | 192-260 (219±24) |
| Rajang Estuary-August, 2016 | 20 | 0-32 (16.3±11.8) | 6.5-8.1 (7.3±0.5) | 24.2-130 (68.4±31.7) | 2.7-4.6 (3.6±0.5) | 0.002-7.0 (1.1±2.2) | 150-357 (245±53) |
| Rajang River -March, 2017 | 2 | 0 | 6.0-6.5 (6.3±0.3) | 116-188 (152±50.9) | 6.3-6.7 (6.5±0.3) | 4.2-8.3 (6.4±2.9) | 126-128 (126 ±1.5) |
| Rajang Estuary- March, 2017 | 13 | 0-30.1 (13.7±12.2) | 6.5-8.2 (7.3±0.6) | 47-327 (151±75) | 4.6-7.6 (6.1±0.7) | 0.004-11.3 (4.2±4.0) | 98-238 (171±42) |
| Maludam-March, 2017 | 9 | 0-20.0 (5.4±6.1) | 3.7-7.6 (4.6±1.4) | 0.4-388 (53.1±121) | 1.1-6.8 (2.7±1.9) | 6.3-23.8 (14.6±6.8) | 0.35*-4.6* (3.6*±1.3*) |
| Sebuyau-March, 2017 | 8 | 0-13.6 (5.4±6.1) | 4.3-7.0 (5.2±1.1) | 0.4-388 (53.1±121) | 1.4-5.9 (3.2±1.9) | 3.0-33.6 (17.6±12.0) | 0.36*-2.1* (1.4*±0.67*) |
| Simunjan-March, 2017 | 6 | 0-0.4 | 4.7-6.3 (5.2±0.6) | 14-481 (135±197) | 1.0-2.6 (1.9±0.7) | 25.8-59.2 (44.2±11.8) | 0.82*-3.1* (2.2*±0.95*) |

*5. As shown in Table 1, the standard deviations for SPM and DOC are rather large. Is it due to the different samples?*

**Reply:** The samples included riverine and estuary samples. So the concentrations of dFe, SPM, and DOC varied between seasons and stations. The precipitation, tide, and sediment composition are critical for the SPM concentration. The SPM increased in the wet season and tide impact area in the Rajang as shown in Fig. 2b, 2g. DOC was higher in the peatland drainage basin, for the peat soil was an abundant source of dissolved organic matter. The DOC concentration was decreased in saline waters in the Rajang Estuary (Supplement: Fig. 2). So the SPM and DOC varied in a wide range, and the standard deviations of total samples are rather large.

**Revised manuscript:** the distributions of DOC in the dry season and wet season were added in Supplement 2.

*6. One or more relative references about "the stronger weathering derived from intensive precipitations" should be added to make the statement of "Considering the limited temperature variation in the tropical zone, the dFe elevation may be related to the stronger weathering derived from intensive precipitations. (Page 9, Line 230-232)" more convincing (needing literature support).*

**Reply:** Thank you for your advice. We reconsidered the statement along with the suggestion of other referees. In Section 4.1, we didn't detect or refer the weathering parameters of this region in different seasons, therefore, the assumption was not convincing. Moreover, the weathering index in the same region can't change significantly within a short time. So we removed this statement, and we attributed this increased dFe concentration to the enhanced transport from the upstream drainage basin (Bhatia et al., 2013). The enhanced precipitation in wet season stimulated the mechanical and chemical weathering in the drainage area, especially in some cropland and soil erosion areas, which is covered commonly in the Rajang drainage basin. So the dFe concentration increased in wet season in the Rajang River.

**Revised manuscript:** Page 10, line 251. As the precipitation enhanced in the wet season, the strong water flow from the upper stream scoured the watershed, carrying the Fe-enriched terrestrial particles to the drainage basin in the wet season (Meade et al., 1985; Taillefert et al., 2000). A great amount of dFe may result from the dissolution of these particle iron originating from mechanical and chemical weathering, which leads to a significant addition of dFe in the wet season (Bhatia et al., 2013).

*7. Check the manuscript carefully. There are several language errors in this paper, for example "supporing" in Page 3 Line 42 and "carring" in Page 9 Line 233. CO 2(In Page 21,Line 569) should be changed to $CO_2$.*

**Reply:** Thank you. We have changed it.

Revised manuscript: Page 3, line 42: change 'supporing' to 'supporting'. Page 22, line 609: change 'CO 2' to '$CO_2$'. "carring" in Page 9 Line 233 removed.

Other grammar and vocabulary errors were also corrected in the revised manuscript and marked in red. For easier understanding, the following sentences were rewritten.

Page 2 line 30: Moreover, the association between dFe and organic matters may result in the conservative distribution of dFe in high salinity waters (salinity>15).

Page 4 line 68: Currently, only limited records on the dFe concentrations were provided in peatland draining rivers (Batchelli et al., 2010; Krachler et al., 2010; Oldham et al., 2017)..

Page 6 line 130: The Maludam River, the majority of which located in the Maludam National Park (the second-largest park in Sarawak), is a pristine river with minor human influences.

Page 8 line 200: The salinity also increased along the water flow pathway    in the Rajang Estuary with an exception in the Rajang tributary

Page 10 line 244: Moreover, there were significantly positive correlations between dFe and DOC in the Sebuyau River and the Simunjan River (Fig. 4c), while the correlation between dFe and DOC in the Maludam River was weak due to an outlier in the high salinity region (S=20.0).

Page 11 line 289: In the wet season, the samples in the Serendeng tributary were collected during a spring tide. Besides, the intensive plantation and agricultural activities in Serendeng tributary modified the soil structure and leached a considerable amount of SPM at flood tide. In Fig. 3a, there was a great dFe and SPM increase at salinity 5-15. Given a similar level in SPM content among the Rajang, Texas River (Jiann et al., 2013) and the Changjiang Estuary (Zhu et al., 2018), we assumed that the dFe enrichment in this special condition may be related to the desorption from the riverine SPM, though we lacked solid supports like the mixing experiment.

Page 13 line 328: The intensive agriculture activities, such as tillage, further enhanced their decomposition, and these activities might improve the mechanical and chemical weathering in the plantation areas, and increased dFe concentration in the Sebuyau River and the Simunjan River as discussed in chapter 4.1.

Page 14 line 381: The thick peatland soils was likely to be the main reason of the high dFe concentration in blackwater rivers, as discussed in the Kiiminkijo River (Heikkinen, 1990), the Tannermoor River (Krachler et al., 2005), the Halladale River (Krachler et al., 2010), the Bebar River (Gastaldo, 2010), and the Taieri River (Hunter, 1983).

**Anonymous Referee #2**

*1. Based on the description in the 2.1 section, the study area covers the drainage basin and estuaries of Rajang River, with major stations locating in the estuaries. So the title should include the "estuaries" inside.*

**Reply:** thank you for your advice. We have changed it.

**Revised manuscript:** As suggested by the referee, we entitle the manuscript to 'Distribution and Flux of Dissolved Iron in Peatland-draining rivers and Estuaries of Sarawak, Malaysian Borneo'.

*2. The description of the study area and Figure 1 are not clear enough to help readers to understand the complex riverine drainage and estuaries. For example:*

*a) Authors should mark the location of Sarawak State in Figure 1a or 1b;*

*b) Normally we don't use bold circle to present riverine drainage basin, it looks like the capital city. Authors should give enough information of different mainstreams and tributaries of discussed riverine basin in Figure 1 to help readers to understand the results and discussion later;*

**Reply:** (a)-(b): thank you for your advice. We have remade the Fig. 1 as the refferee's advice.

**Revised manuscript:** We remade Fig. 1a 1b. We marked the location of Sarawak State in Fig. 1a and repainted Fig. 1b to distinguish the complex river systems, especially the mainstreams and tributaries of some black water rivers.

*c) In line106 (page 5), authors mention that "The Igan tributary is the main outlet for freshwater (of Rajang River)". However, the survey carried out in the wet season (March 2017) only had one station in this tributary. There was only one sampling station in the main stream of Rajang River in wet season too. Authors should explain the sampling strategy between two voyages. Otherwise the discussion to compare the results between two seasons are hard to understand due to different stations' coverage.*

**Reply:** Igan as the major outlet for the freshwater of Rajang River. We collected 6 samples along the upstream to the estuary of this tributary to figure out the dFe distribution in August 2016. However in March 2017, there were some sampling incidences result from limited time and mechanical failure. There were only 2 stations in Igan tributary. Because the sampling ship can't pass the shallow water near the estuary owing to the huge depositions of mud and sludge. In March 2017, there were also limited samples in the Rajang River sample. Because our sampling ship can't access to the upper basin due to the high current velocity. The dFe concentration in the freshwater near the Sibu was higher in the wet season (2 samples), although we failed to compare the sufficient samples in the upstream of the Rajang River. For the Rajang Estuary, the samples in the different cruises can't keep the same path, but we tried to cover the major tributaries of the Rajang Estuary in the limited sampling time.

**Revised manuscript:** Page 6, line 144: In March 2017, we failed to collect samples in the upstream of Rajang and the saline samples in Igan tributary mainly due to the shallow water depth and strong

current. However, three blackwater rivers, as aforementioned, were included in the cruise.

*d) In Figure 3, authors don't explain how to separate the stations in the Rajang Riverine basin into three parts of "Rajang, Igan and Serendeng"? And where is "Serendeng" in Figure 1? How do authors separate riverine stations and estuary stations between two sampling voyages? According to the variations of salinity or using Sibu city as the boundary (Page 5 Line 101-102)? Relevant information is very important for the understanding of results of different average concentrations and behavior of dFe between dry and wet seasons.*

**Reply:** From the Sibu city, the Rajang River is separated into several tributaries. The Igan tributary is the major outlet of Rajang. The tributary Hulu Serendeng is then separated into Paloh and Lassa. And tributary Belawai and Rajang are in the south of the Rajang Estuary. In Fig. 3, the stations in tributary Igan were shown as circle '●/○'. The stations in tributary Paloh and Lassa were shown as '■/□' and noted as 'Serendeng'. The stations in tributary Belawai and Rajang were shown as '▲/△' and noted as 'Rajang'. There was only 1 station in Belawai tributary in August 2016, which was closed to the Rajang tributary, and with similar tidal impact and watershed condition. Therefore we combined the results of these two tributaries in Fig. 3. In order to clarify the figure, we supply the notes of the symbol points of Serendeng and Rajang in Fig. 3 and mark the tributaries in Fig. 2. The descriptions of the tributaries were also supplied in Section 3.2.

**Revised manuscript:**

Page 5, line 103: Sibu city is assumed to be the boundary line of the Rajang drainage basin and the Rajang Estuary according to the physiographic condition (Staub et al., 2000; Staub and Esterle, 1993), and the saltwater intrusion could reach the downstream to the city (Jiang et al., 2019).

Page 5, line 109: There are several tributaries for the Rajang River in the estuary, including Igan, Hulu Serendeng (further separated into two tributaries: Paloh and Lassa), Belawai and Rajang.

Page 9, line 217: We adopted Sibu as the separation of the Rajang River and the Rajang Estuary.

Page 9 line 225: The sites in tributary Paloh and Lassa were combined as Serendeng tributary, and the tributary Belawai and Rajang were combined as Rajang tributary.

Figure 3: We supply the notes of the symbol points of Serendeng and Rajang. The figure title: Figure 3: Dissolved iron (dFe) correlation with salinity (S) (a), suspended particulate matter (SPM) (b), dissolved organic carbon (DOC) (c), dissolved oxygen (DO) (d), and pH (e) in Rajang estuary. The solid lines were the linear regressions between dFe and other factors, and the colors of the lines were coincident with the data points in different salinity range. Serendeng included the station in tributary Paloh and Lassa, and Rajang included the station in tributary Belawai and Rajang.

Figure 2: We mark the tributaries of the Rajang in Figure 2.

*e) Since the behavior of dFe is strongly affected by the SPM based on the discussion, authors should give the information of sediment loads of Rajang River and the other three blackwater rivers.*

**Reply:** Thank you. In Rajang River, we invoked the result of Staub and Gastaldo (2003) in Page 72, that 'the Rajang River drainage basin provided ~ 30 million metric tons of sediment annually', and this information was in Page 6 line 129 in the manuscript. However, for the three balck water rivers,

we didn't find any references and sufficient data concerning about their sediment loadings.

Moreover, as the reviewer's advice, we also supply some informations about the the grain size of sediment in these river drainage basins. We think that it may be critical for sediment resuspension progress. The grain sizes in the Rajang Estuary was much lower than the Rajang River. This further proved our statement in Section 4.2, that the resuspension of bottom sediments may be an important dFe source in the Rajang Estuary, especially in the wet season and after the spring tide. The other three rivers are typical blackwater rivers, with the rich peatland drainage basin. The sediments of black water rivers were finer-grained, and mainly from woody material. As the source of sediment and terrestrial organic carbon, peatland may also be the critical source of dFe as we discussed in Section 4.3.

**Revised manuscript:**

Page 5 line 118: The Rajang riverine freshwater drains the mineral soil, so the mean grain sizes of the sediment were much coarser than the Rajang Estuary, where peatland is dominant in the delta region (Wu et al., 2019).

Page 6 line 136: The grain size of sediments in blackwater rivers was much lower, and received more woody material than that of the Rajang River (Wu et al., 2019).

*f) What's the water depth of the Rajang River? Since only surface samples were collected and authors explain the difference of dFe behavior between two seasons based on the resuspension and adsorption/desorption on the surface of particles. It's better to give the basic information of Rajang River.*

**Reply:** thank you for your advice. Water depth information: As the reviewer's good advice, the information of water depths, along with the flow velocity, was added in Section 2.1 to clearly describe the hydrodynamic conditions in the Rajang River.

**Revised manuscript:** Page 5 line 96: The Rajang River was around 5-10 m and 8-20 m deep in the dry season and wet season respectively, whereas the tributaries <5 m deep. The flow velocity ranged from 0.2-0.6 m s$^{-1}$ and 0.8-1.2 m s$^{-1}$ at mainstream in the dry season and the wet season respectively (Tawan et al., 2019).

*3. Authors prove the robust of dFe measurement by the international intercalibration experiments results using SAFe D1 and SLEW-3. dFe measurements using isotope dilution ICP-MS method have good precision and accuracy. However, the samples collected in the Rajang River and the other three blackwater rivers have high content of DOM which is quite different with the open ocean and coastal region. What's species of dFe can be measured using solid extraction method depends on the selection of resin and the composition of organic ligands in the riverine and estuarine samples. Authors should give detailed information of resin type in section 2.3 and give clear definition of what's the speciation of measured "dFe" and what's the difference between measured "dFe" and total dissolved Fe (measured after digestion) in the section of results and/or discussion.*

**Reply:** thank you for your advice. The samples were acidified to pH 1.7-2.0 to release metal from organic ligands to inorganic form. Then the samples were preprocessed using the single batch nitrilotriacetate (NTA)-type chelating resin. The dissolved Fe$^{3+}$ can be recovered quantitatively at

pH<2 after the oxidization of $Fe^{2+}$ to $Fe^{3+}$ by the addition of $H_2O_2$. So, the measured dFe was the entire dissolved Fe in the samples. Considering the low concentration of dFe near the coastal area, we used some intercalibration samples, like SAFe D1 and SLEW-3, to confirm that our detection accuracy was still reliable in trace level, which is critical in trace metal detection. In order to keep the integrity of the section result/discussion, we add the supplement in sections 2.2 and 2.3. In section 2.2, we declared that the organic complex Fe was transformed into inorganic form by acidized to pH 1.7-2.0. In section 2.3, we explained that the dissolved Fe was oxidized to $Fe^{3+}$ and complexed with NTA resin.

**Revised manuscript:**

Page 7 line 154: The samples then thawed at room temperature in the clean laboratory and acidified with ultrapure HCl to pH 1.7 in an ultra-clean lab to transform and preserve metal Fe in soluble inorganic form (Lee et al., 2011).

Page 7 line 162: The acidified samples were preprocessed by the single batch nitrilotriacetate (NTA)-type chelating resin (Qiagen Inc., Valencia, CA). The dissolved Fe can be recovered quantitatively after the oxidization of $Fe^{2+}$ to $Fe^{3+}$ by the addition of $H_2O_2$ (Lee et al., 2011).

*4. Page 8 Line 182-190: Authors discussed the distributions of SPM, DO and pH in the two seasons of the Rajang River and Estuary. However, it's hard to follow because we don't know locations of Serendeng tributary and Rajang tributary. The difference of color bar in Figure 2b and 2c is hard to recognize.*

**Reply:** thank you for your advice. We remade Fig. 2.

**Revised manuscript:**

We made some corrections in Fig. 2. The tributaries were marked in the figures including Igan, Paloh, Lassa, Belawai, and Rajang. The color bar in Fig. 2b and 2c was clarified to differentiate the variation of SPM and DO in August 2016. The notes of the Serendeng and Rajang tributary was added in Fig. 2 and Fig. 3 and discussed in 2(d).

Page 8 line 199: remove 'In Serendeng tributary, some high salinity samples inside the river mouth in the wet season were found.' add 'The salinity also increased along the transportation in the Rajang Estuary with an exception Rajang tributary.'

*5. (a)Section 3.2: As mentioned in the comments 2-d, I don't know how do authors separate the riverine and estuarine stations between dry and wet seasons, the data of concentration ranges and averages are hard to understand due to different sample coverage and data number.*

**Reply:** Thank you. We referred to the result of Staub and Gastaldo (2003), which separated the wet season and dry season according to the discharge from the Rajang River drainage basin. The precipitation also indicated that the rainfall decreased significantly in April and October of inter-monsoon period of Sarawak (Sa'adi et al., 2017). Therefore, August 2016 and March 2017 were adopted as the dry season and the wet season respectively in this paper.

The Rajang River and the Rajang Estuary were separated according to the physiographic condition and was located in Sibu city. Other results also indicated that the saltwater of the South China Sea

can intrude the Rajang Estuary reaching the Sibu city (Jiang et al., 2019; Müller-Dum et al., 2019). Therefore, we referred to this consensus that stations in the upstream before Sibu were the Rajang riverine station, and the downstream stations after Sibu were the Rajang Estuary stations. In order to reduce the misunderstandings of the concentration range and mean of the riverine and estuary, we separated the stations into the Rajang River and the Rajang Estuary (Sibu as the boundary) in Table 1, and make the results consistent between the Table 1 and Section 3.2.

*(b) Ordinate of the iron concentration in figure 3 using two different scales, what's the scientific hypothesis of this separation? Using salinity >15 and <15 as the separation boundary? If this is true, authors should differentiate the salinity using the same boundary in Figure 2.*

**Reply:** thank you for your advice. From Fig. 3a, there was the intensive removal of dFe within salinity=15 in the Rajang Estuary, especially in August 2016. For Fig. 3b-3e, the same magnitude of iron concentration ordinate can't specify the relationship of dFe with SPM, DOC, DO, and pH. In order to reveal the correlation between dFe and other factors (SPM, DOC, DO, and pH), we amplify the low salinity zone (S<15), so the distinct relationships of dFe concentration with other factors before/after great removal were much clearer. We take the full consideration of the comments and added the isosalinity line (S=15) linear interpolated from salinity in Fig. 2a, 2f to display the boundary of the salinity in different seasons.

**Revised manuscript:**

Page 9 line 218: The dFe concentrations in the Rajang River ranged from 3.3 to 7.3 μmol L$^{-1}$ (mean: 5.5±1.7 μmol L$^{-1}$) in August 2016, and ranged from 4.2 to 8.3 μmol L$^{-1}$ (mean: 6.4±2.9 μmol L$^{-1}$) in March 2017.

Figure 2a, 2f: we added the isosalinity line (S=15) linear interpolated from salinity

Figure 2 title: Figure 2. Spatial distributions of salinity (a) (f), suspended particulate matter (SPM) (b) (g), dissolved oxygen (DO) (c) (h), pH (d) (i), dissolved iron (dFe) (e) (j) in the Rajang River in August 2016 and March 2017. The red solid line is the isosalinity line (S=15) linear interpolated from S in this region.

*6. Section 4.1 page 9 line 231-232: I can't agree with the statement of "dFe elevation may be related to the stronger weathering derived from intensive precipitation". Weathering index in the same region between dry and wet season can't change significantly. The elevation of dFe in the wet season might be caused by the dissolution of weathering product in the riverine drainage basin due to the runoff variation and precipitation, and might also due to the resuspension of bottom sediments.*

**Reply:** We agree with the reviewer's suggestion in Section 4.1, that the high concentration of dFe in wet season was minor related to the weathering. We didn't detect or refer the weathering parameters of this region in different seasons, therefore, the assumption was not convincing. So we removed this statement and attributed this increased dFe concentration to the dissolution of weathering products from the upstream drainage basin based on the referee's suggestion. The enhanced precipitation in wet season stimulated the mechanical and chemical weathering in the drainage area, especially in some cropland and soil erosion areas, which is common in the Rajang

drainage basin. So the dFe dissolved from the weathering product increased in the wet season at the Rajang River.

For the reviewer's suggestion about resuspension of bottom sediments, it might be more important in the Rajang Estuary. There was a great addition of dFe in the Rajang Estuary in wet season especially in Serendeng tributary, where the samples were collected after the spring tide. The bottom samples indicated that the bottom dFe concentration was about 4 times of surface dFe concentration, and the SPM was elevated significantly in this tributary after the spring tide. So we also add the resuspension impact in Section 4.2.

**Revised manuscript:**

Page 6 line 148: The bottom samples were collected using a pre-cleaned 5 L Teflon-coated Niskin-X bottle hung on a nylon rope. Due to the limited sampling time and condition, only 3 bottom samples in August 2016 and 1 bottom sample in March 2017 were collected.

Page 10, line 251: As the precipitation elevated in the wet season, the rapid water mass from the upper stream scoured the watershed, carrying the Fe-enriched terrestrial particles to the drainage basin in the wet season (Meade et al., 1985; Taillefert et al., 2000). A great deal of dFe may be dissolved from these particle iron originating from mechanical and chemical weathering, which is a significant addition of dFe in wet season (Bhatia et al., 2013).

Page 12 line 296: In addition, the limited bottom samples in the Rajang Estuary also revealed that the dFe addition within salinity 5-15 in the wet season might also result from the resuspension of bottom sediments, for the bottom dFe concentration was much higher than the surface dFe concentration.

*7. Authors using correlations of dFe with other parameters (e.g. SPM, DO etc) to discuss the behavior of dFe in the riverine and estuary region and between two seasons. Discussion is relatively superficial and need to add some solid evidence.*
**Reply:** It's our negligence and we are sorry about this. Indeed, the analysis in the discussion was relatively superficial due to the lack of sufficient data to support our assumptions. So it's necessary to set up the subsequent studies to investigate the complex biogeochemical mechanisms in the future works.

These two cruises were the first attempt to carry out the related research, so the background result was rare to see. So we encountered a lot of incidents in the sampling period. For example, we failed to perform some critical experiments during the cruise due to the intensive sampling schedule, like the mixing experiment, which could validate some assumptions in our discussions. In the wet season, the flow velocity increased extremely a lot. So our sampling ship was difficult to arrive at some sites, and the original schedule was delayed in the Rajang River and the Rajang Estuary. Even though, we tried to collect samples in major tributaries and estuaries along with other hydrographic properties to figure out the dFe distribution and speculate it's impact factors in these peatland-draining rivers in Sarawak.

In order to verify our assumptions, we introduced a lot of references and our published results in other rivers and estuaries. During the discussions, we can see that the dFe distribution was affected greatly by SPM in different conditions. The resuspension of SPM was a great addition of dFe in the Rajang River and the Rajang Estuary. We take this assumption according to the references and the

limited bottom samples in our cruises. The comparison of dFe concentrations in surface and bottom samples was shown as Fig. R1. The difference of the dFe concentration between the surface and bottom was not significant in the Rajang River. In the Rajang Estuary, the dFe concentration in the bottom was much higher than the surface, so we attributed the addition of dFe in this region in the wet season to the resuspension of sediments. But we didn't include the bottom results in the manuscript owing to the limited samples.

Moreover, we also attributed the addition of dFe to desorption from SPM according to the result in the Changjiang Estuary. The dFe was increased a lot in the turbidity maximum zone (Fig. R2), and the mixing experiment suggested that the desorption process dominated in the simulation period (Fig. R3). Combined with other references in similar conditions in rivers, our assumption could make sense.

According to the correlation between dFe with other parameters, we can confirm the impact factors on the distribution of dFe, and explore the possible biogeochemical process of dFe in the rivers referring to the results in similar river systems. In this manuscript, we just performed the distribution and flux of dFe in these peatland-draining rivers and suggested some possible impact factors. To verify these biogeochemical mechanisms, more samples and simulation experiments are needed in our future work. Thanks for the referee's good evaluation and kind suggestion, which provide the critical guide for our later researches.

[Figure]

Figure R1: The comparison of dFe concentration in surface and bottom samples in two cruises.

[Figure]

Figure R2: the dFe concentration in the Changjiang Estuary (Zhu et al., 2018)

[Figure]

Figure R3: dFe results of leaching experiment versus (a) time and (b) SPM gradient, mixing experiment of filtered seawater (FSW) mixed with (Zhu et al., 2018).

**Other comment:**

1. In the former manuscript, the Fig.5 appeared later than Fig.6, so we reorder the sequence of two figures, and make the correations in this version as follow: Page 12 Line 316: from Fig. 6a to Fig. 5a; Page 13 Line 342: from Fig. 6b to Fig. 5b; Page 14 Line 358: from Fig. 5 to Fig. 6; Page 14 Line 383: from Fig. 5b to Fig. 6b.

2. In the former manuscript, 'the Rajang freshwater' and 'the Rjang River' were both used to described samples in the Rajang River endmember, which was confused to understand the concentration and distribution of dFe. In the revised version, we uniform the expression of 'the Rajang River' to indicate the Rajang freshwater endmamber, which located in the upstream of Sibu city.

3. In the revised Fig. 6, the distribution of the rivers was figured in Fig. 6c to separate the tropical rivers with others.

**References**

Bhatia, M. P., Kujawinski, E. B., Das, S. B., Breier, C. F., Henderson, P. B., and Charette, M. A.: Erratum: Greenland meltwater as a significant and potentially bioavailable source of iron to the ocean, Nature Geoscience, 6, 503-503, 2013.

Jiang, S., Müller, M., Jin, J., Wu, Y., Zhu, K., Zhang, G., Mujahid, A., Rixen, T., Muhamad, M. F., Sia, E. S. A., Jang, F. H. A., and Zhang, J.: Dissolved inorganic nitrogen in a tropical estuary in Malaysia: transport and transformation, Biogeosciences, 16, 2821-2836, 2019.

Klunder, M. B., Bauch, D., Laan, P., de Baar, H. J. W., van Heuven, S., and Ober, S.: Dissolved iron in the Arctic shelf seas and surface waters of the central Arctic Ocean: Impact of Arctic river water and ice-melt, Journal of Geophysical Research: Oceans, 117, 2012.

Müller-Dum, D., Warneke, T., Rixen, T., Müller, M., Baum, A., Christodoulou, A., Oakes, J., Eyre, B. D., and Notholt, J.: Impact of peatlands on carbon dioxide (CO2) emissions from the Rajang River and Estuary, Malaysia, Biogeosciences, 16, 17-32, 2019.

Zhu, X., Zhang, R., Wu, Y., Zhu, J., Bao, D., and Zhang, J.: The remobilization and removal of Fe in estuary-A case study in the Changjiang Estuary, China, Journal of Geophysical Research Oceans, 2018.

---

## Author Response (AR2)

Dear Editors,

We are very pleased to know that our manuscript is acceptable for publication in Biogeosciences with the minor revision.

In the recent version, we have substantially revised our manuscript after reading the comments, provided by the Associate Editor, and we employed an English-language editing service, AJE, to polish our word and grammar. On the other hand, we corrected some information in References list based on the reference types of Biogeosciences.

We would like to express our appreciations to you and the referees for suggesting how to improve our paper.

Best regards,
Xiaohui Zhang